# Characterization of the Gut Microbiota of Mackerel Icefish, *Champsocephalus gunnari*

Hokyung Song [1,2], Seungyeon Lee [1,3] , Dong-Won Han [1] and Jin-Hyoung Kim [1,3,*]

1   Division of Life Sciences, Korea Polar Research Institute, Incheon 21990, Republic of Korea
2   Subtropical/Tropical Organism Gene Bank, Jeju National University, Jeju 63243, Republic of Korea
3   Polar Science, University of Science and Technology, Daejeon 34113, Republic of Korea
*   Correspondence: kimjh@kopri.re.kr

**Abstract:** The gut microbiome of Antarctic fish species has rarely been studied due to difficulties in obtaining samples. The mackerel icefish, *Champsocephalus gunnari*, primary feeds on krill and is one of the key species in the food web of the Southern Ocean. In this study, we characterized the gut microbiota of *C. gunnari* by sequencing the V3–V4 region of the bacterial 16S rRNA gene based on the Illumina MiSeq sequencing platform. We collected three types of samples: (1) whole intestine, (2) intestinal wall, and (3) intestinal content. The results showed no significant difference in the alpha diversity between different sample types. However, the microbial community composition of intestinal wall samples was distinct from other sample types. The relative abundance of *Photobacterium* was higher in intestinal content compared with the walls, which could be due to their chitinolytic activity. In contrast, potential pathogens such as *Escherichia*, *Shigella*, and *Pseudomonas* relatively more abundant in the intestinal wall compared with the intestinal contents. Unlike the gut microbiome of other marine fish species, *Vibrio* and *Lactobacillus* were nearly absent in the gut microbiome of *C. gunnari*. Functional gene profile of the gut microbiome predicted by PICRUSt2 showed higher relative abundance of genes related to biodegradation of nutrients in intestinal content. In contrast, the relative abundance of genes related to biosynthesis of important metabolites, such as menaquinols, was higher in intestinal wall. The difference in the microbial community structure of intestinal wall and intestinal content found in our study supports niche separation in the gut environment and emphasizes the importance of collecting intestinal wall samples in addition to intestinal content samples to understand the full picture of gut microbiome. This is the first time that the gut microbiome of mackerel icefish has been characterized using next-generation sequencing.

**Keywords:** gut microbiome; Antarctic fish; mackerel icefish; *Champsocephalus gunnari*; high-throughput sequencing

## 1. Introduction

The gut microbiome of vertebrates has been extensively studied because they play important roles in host health and survival [1]. In humans, beneficial gut microbes such as *Bifidobacterium*, *Prevotella*, and *Faecalibacterium* aid the digestion of non-digestible fibers, producing short-chain fatty acids which are important metabolites in host energy metabolism and immunity [2,3]. On the other hand, the gut could serve as a reservoir for pathogens such as *Escherichia*, and *Shigella*, which cause diarrhea, and other bacterial species which cause more severe diseases [4]. Depending on the host age, geography, physiological conditions, and diets, gut microbial compositions change simultaneously [5–7].

Research on the fish gut microbiome has increased recently in parallel with the expansion of the aquaculture industry [8]. The gut microbiome of commercially valuable species such as salmon [9–14] or model species such as zebrafish [15–18] have been well characterized. In contrast, the gut microbiome of wild fish has scarcely been studied although they are one of the key members in the ecosystem. More specifically, due to limited accessibility,

the gut microbiome polar fish species are largely unknown. Among Antarctic fish species, the gut microbiota of *Notothenia coriiceps, Chaenocephalus aceratus* [19], *Trematomus bernacchii, Chionodraco hamatus, Gymnodraco acuticeps, Pagothenia borchgrevinki* [20], and some Antarctic lanternfish species [21] have been characterized to date.

In this study, we applied high-throughput sequencing technique to identify the gut microbiome of mackerel icefish, *Champsocephalus gunnari*. *C. gunnari* spend most of their lifetime in the cold seawater of the Southern Ocean; therefore, their physiology is distinct from non-polar fish species. For example, because they live in cold water where the amount of dissolved oxygen is high, they have transparent blood, lacking hemoglobin [22,23]. Additionally, the total lipid contents in *C. gunnari* are higher than that of non-polar fish species, which could be helpful for the maintenance of body temperature [24]. Due to their unique physiological characteristics, we expected *C. gunnari* to have a distinct gut microbiome. To the best of our knowledge, the gut microbial composition of *C. gunnari* has not yet been studied, and only the gut microbial composition of their relatives, *Chionodraco hamatus*, which belongs to the same family (Channichthyidae), has been studied [20]. In this study, we collected three types of samples (whole intestine, intestinal wall, and intestinal content) to distinguish temporally abundant bacterial groups and persistent bacterial groups. To address the possible role of gut microbes, we have also predicted functional gene profiles based on bacterial community structure using PICRUSt2 software [25].

## 2. Materials and Methods

### 2.1. Sample Collection

The mackerel icefish, *Champsocephalus gunnari* (Supplementary Figure S1), were obtained from Jeong Il Corporation, a krill fishery. The wild *C. gunnari* were caught as a by-catch species in the Subarea 48.1 and 48.2 of the CAMLR (Convention on the Conservation of Antarctic Marine Living Resources) Convention area [26] from November 2020 to February 2021. All samples were stored in a freezer ($-20$ °C) immediately after collection and were transported in frozen state ($-20$ °C) to the Korea Polar Research Institute within 6 months. Eight individuals of similar size and weight from the same age cohort were selected for this study (Table 1). Fish samples were defrosted at 4 °C for 12 h before dissection. Before dissecting the fish, we wiped the ventral body surface with a paper towel to remove excess mucus. The surfaces of each fish were treated with 70% ethanol for sanitization and dried with a paper towel. Dissection tools were sanitized with 70% ethanol and flame-sterilized every time we changed the fish individuals, to prevent cross-contamination between samples. The digestive organs (from esophagus to vent) were pulled out from the fish, avoiding rupture of the gallbladder. The intestines from each individual fish were aseptically collected, and adipose tissues attached to the intestine were removed using forceps.

**Table 1.** Physiological details of the collected samples.

| Fish ID | Sample ID * | Total Length (cm) | Standard Length (cm) ** | Wet Weight (g) | Sex (M/F) | Gut Weight (g) |
|---|---|---|---|---|---|---|
| F1 | WI1 | 37.5 | 34 | 431.2 | F | 9.37 |
| F2 | WI2 | 40 | 36 | 515.3 | M | 12.6 |
| F3 | WI3 | 38.5 | 34.2 | 443.56 | M | 12.4 |
| F4 | WI4 | 38.5 | 34.2 | 503.21 | F | 13.38 |
| F5 | IW1, IC1 | 40 | 36 | 512.53 | M | 12.65 |
| F6 | IW2, IC2 | 37.5 | 33.7 | 410.73 | F | 9.28 |
| F7 | IW3, IC3 | 36.5 | 33 | 453.16 | F | 13.11 |
| F8 | IW4, IC4 | 37 | 34 | 429.01 | F | 11.11 |

* WI: whole intestine; IW: intestinal wall; IC: intestinal content. ** Standard length: (total length) $-$ (tail length).

The total intestine from the caeca to the anus (proximal and distal intestine), including the intestinal contents and wall, was collected into individual sterile 50 mL tubes from four

out of eight individuals. From the other four individuals, we obtained intestinal contents and intestinal walls separately, by squeezing out the contents using sterile spreaders (Supplementary Figure S2). Intestinal contents from the four individuals were collected into sterile 50 mL tubes. We collected the intestinal walls in sterile 50 mL tubes filled with 15 mL of phosphate-buffered saline (pH 7.4) and rinsed the samples by inverting the tube 10 times. The intestinal wall samples were then homogenized at approximately 13,400 rpm (wheel scale of 4) for 30 s three times (5 s rest in each interval) using a T-10 basic ULTRA-TURRAX® homogenizer with an S10N-8G dispersal tool.

### 2.2. DNA Extraction and 16S rRNA Gene Amplicon Sequencing

DNA was extracted using the DNeasy Powersoil Pro Kit (Qiagen, Seoul, Republic of Korea, Cat No. 47016) from 300 μL of the samples, following the manufacturer's protocol. Sample DNA was sent to Celemics (Seoul, Republic of Korea) for 16S rRNA gene amplicon sequencing. Using the forward primer 341F (5′-CCTACGGGNGGCWGCAG-3′) and the reverse primer 805R (5′-GACTACHVGGGTATCTAATCC-3′), the V3–V4 region of 16S rRNA gene was amplified. CLM Polymerase (Celemics, Seoul, Republic of Korea) was used for PCRs, with the following PCR conditions: (1) initial denaturation (95 °C, 5 min); (2) 10 cycles of denaturation (95 °C, 30 s), annealing (62 °C, 30 s), extension (72 °C, 30 s); and (3) final extension (72 °C, 5 min). Before pooling, samples were indexed using the Nextera XT Index Kit (Illumina, Seoul, Republic of Korea, Cat No. FC-B1-1001). Paired-end Illumina MiSeq sequencing was performed using the MiSeq Reagent Kit v3 (2 × 300 bp) (Illumina, Cat No. MS-102-3003). One of the WI samples could not be sequenced due to the low amount of DNA.

### 2.3. Bioinformatics

Paired-end sequences were assembled using PANDAseq software with a minimum overlap of 10 bp [27]. Sequences were further processed using Mothur v. 1.39.0 following the MiSeq SOP (https://mothur.org/wiki/miseq_sop/, assessed on 1 July 2021). Sequences were aligned and classified based on the Silva v. 138 database [28]. Sequence reads that were not properly aligned on the expected range were removed using the "screen.seqs" command. Chimeric sequences were removed using the "chimera.vsearch" command [29]. Sequences assigned as "Eukaroyta", "unknown", "Chloroplast", and "Mitochondria" were removed. Operational taxonomic units (OTUs) were defined with 97% sequence similarity. PICRUSt2 [25] was utilized to infer the functional gene composition of each sample. Sequence reads were normalized with 17,011 reads per sample prior to applying PICRUSt2 and performing statistical tests. Based on EC number abundances, MetaCyc pathway abundances were inferred.

### 2.4. Statistical Analysis

Analysis of variance (ANOVA) was performed to compare alpha-diversity between sample groups. Bray–Curtis dissimilarities between samples were calculated based on the square-root-transformed OTU table, and were visualized through the non-metric multidimensional (nMDS) plot. Analysis of similarity (ANOSIM) was performed to test whether the microbial community structures differed by sample type or sex group. We used Primer v. 6 software [30] to generate the nMDS plot and to perform ANOSIM tests. linear discriminant analysis effect size (LEfSe) analysis was performed on the Huttenhower Galaxy Server (http://huttenhower.sph.harvard.edu/galaxy/, assessed on 24 September 2022) to determine differentially abundant genera and metabolic pathways in the intestinal wall and intestinal content samples. Genera with fewer than 10 reads were not included in the LEfSe analysis. *p*-values from the LEfSe analysis were not adjusted because the number of samples used in this study was minimal.

## 3. Result and Discussion

After quality filtering, we obtained 717,468 reads in total, ranging from 17,011 to 90,417 reads per sample. There were no significant differences in the alpha diversities (number of OTUs and Shannon diversity) by sample types (Figure 1). Figure 2 shows the nMDS plot of the studied samples grouped by sample type. The ANOSIM result showed significant differences in the microbial community of the intestinal content and that of intestinal wall (R = 0.625, *p* = 0.029). The differences in the microbial community of the whole intestine and that of intestinal wall were marginally significant (R = 0.463, *p* = 0.057). There were no significant differences between the microbial community of the whole intestine and that of intestinal content (R = 0, *p* = 0.429). Sex-related differences were not observed (Supplementary Figure S3, ANOSIM global R = −0.061, *p* = 0.621).

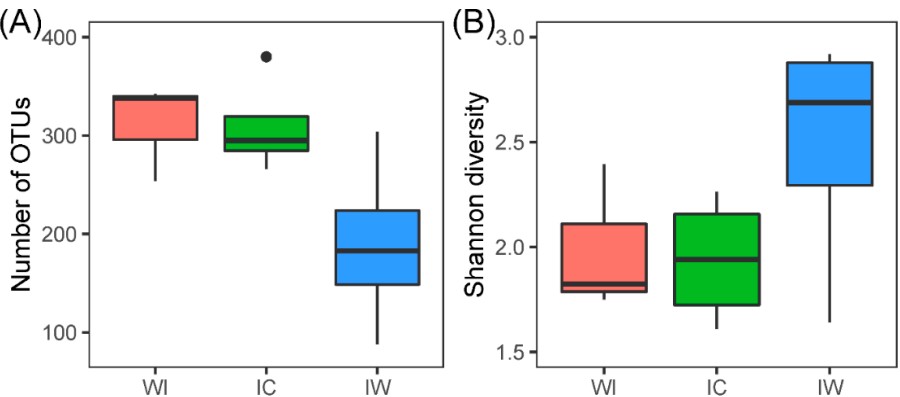

**Figure 1.** (**A**) Number of bacterial OTUs in each sample type. (**B**) Shannon diversity of the bacterial communities in each sample type. There were no significant differences in the alpha diversities by sample type. WI, whole intestine (n = 3); IW, intestinal wall (n = 4); IC, intestinal content (n = 4).

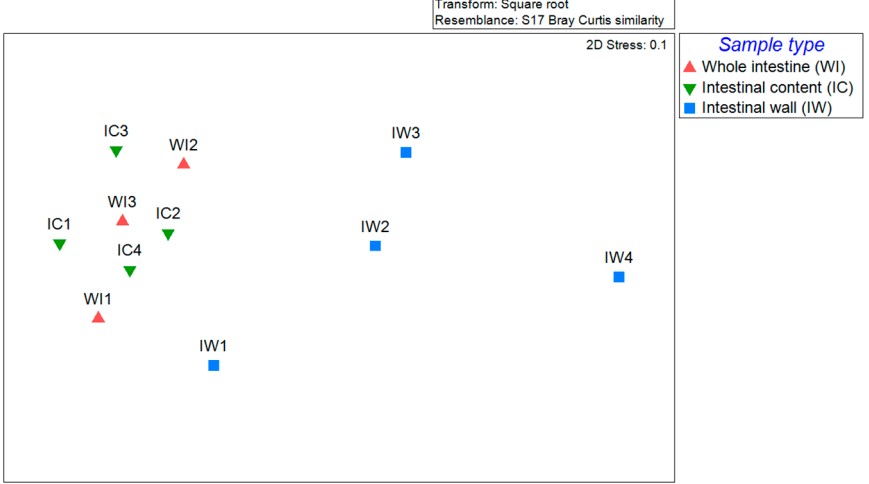

**Figure 2.** Non-metric multidimensional (nMDS) plot showing Bray–Curtis distances between the gut microbiome samples of wild *C. gunnari*.

At phylum level, the gut microbiota of *C. gunnari* was dominated by Firmicutes, followed by Proteobacteria (Figure S4), which corresponds with previous studies on the fish gut microbiome. Kim et al. [31] analyzed the gut microbiome of 227 individual fish species and found that Proteobacteria and Firmicutes dominated. Song et al. [20] studied the gut microbiome of four different Antarctic fish species and reported the dominance of Firmicutes and Proteobacteria in the gut microbiome of *Chionodraco hamatus* which belongs to the same family as that of *C. gunnari* (Channichthyidae).

At genus level, unclassified Clostridiaceae was the most dominant group on average, followed by *Photobacterium*, *Paeniclostridium*, and *Mycoplasma* (Figure 3). *Lactobacillus* and *Vibrio*, both of which are commonly found in the gut of fish [32,33], were nearly absent from most of the studied samples. Clostridiaceae is commonly found in the gut of mammals and fish [34]. Clostridiaceae includes human pathogens, such as *Clostridium difficile*, which cause pseudomembranous colitis, and *Clostridium perfringens*, which cause clostridial necrotizing enteritis [35]. However, some of the others belonging to Clostridiaceae are beneficial to their host, e.g., *Clostridium butyricum* can produce butyrate, which plays an important role in energy metabolism and in the immune system [36]. However, relatively little is known about the role of Clostridiaceae in the fish gut, requiring further investigation.

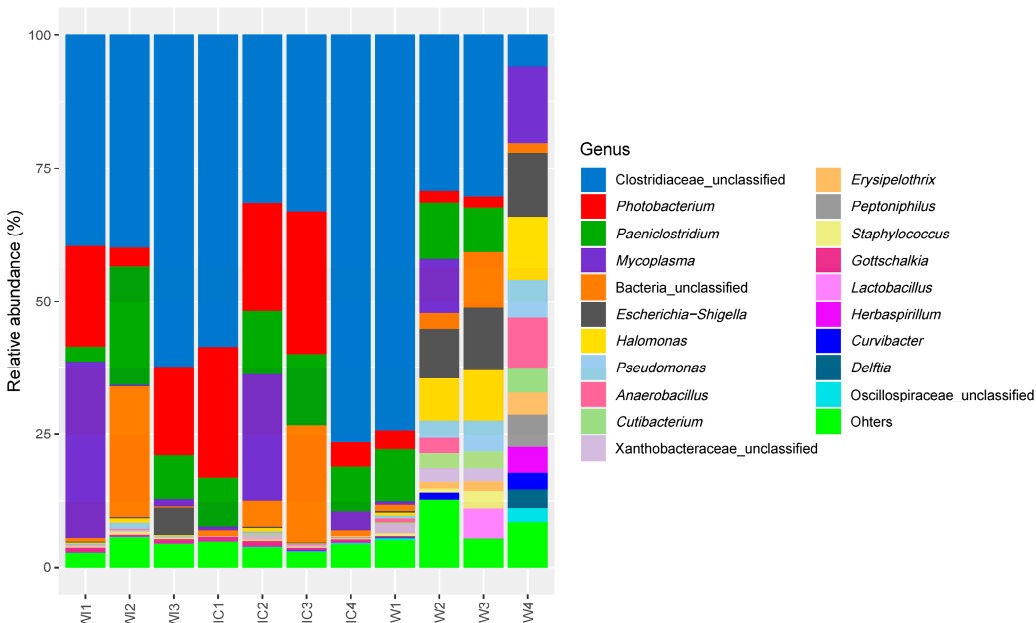

**Figure 3.** Genus composition of the *C. gunnari* gut microbiome. WI, whole intestine; IW, intestinal wall; IC, intestinal content.

*Photobacterium* have been suggested as mutualistic bacteria due to their chitinolytic activity [8,37]. The relative abundance of *Photobacterium* in intestinal content was 18.9% on average, and was significantly higher than that in the wall based on the LEfSe analysis (Figure 4). *C. gunnari* primary feeds on Antarctic krill, which have a chitin-rich shell [38]. *Photobacterium* could possibly aid in the digestion of the shells of krill. Higher relative abundance of *Photobacterium* in intestinal wall compared with intestinal content suggests that *Photobacterium* could be present transiently in the *C. gunnari* intestine when they eat krill.

*Paeniclostridium* has recently been proposed as a new genus [39]; therefore, species belonging to *Paeniclostridium* have not been studied extensively. The only exception is *Paeniclostridium sordellii*, which was first isolated in 1922, then renamed and reclassified afterward to its current phylogeny [40]. *Paeniclostridium sordellii* produce toxins and are pathogenic to mammals, including humans, causing gas gangrene (myonecrosis), sepsis, fatal toxic shock syndrome, and enterocolitis [41]. However, their effects on fish species have not been studied sufficiently. They were one of the dominant species in our study; thus, it would be necessary to further investigate their association with Antarctic fish species because they have detrimental effects on other animal species.

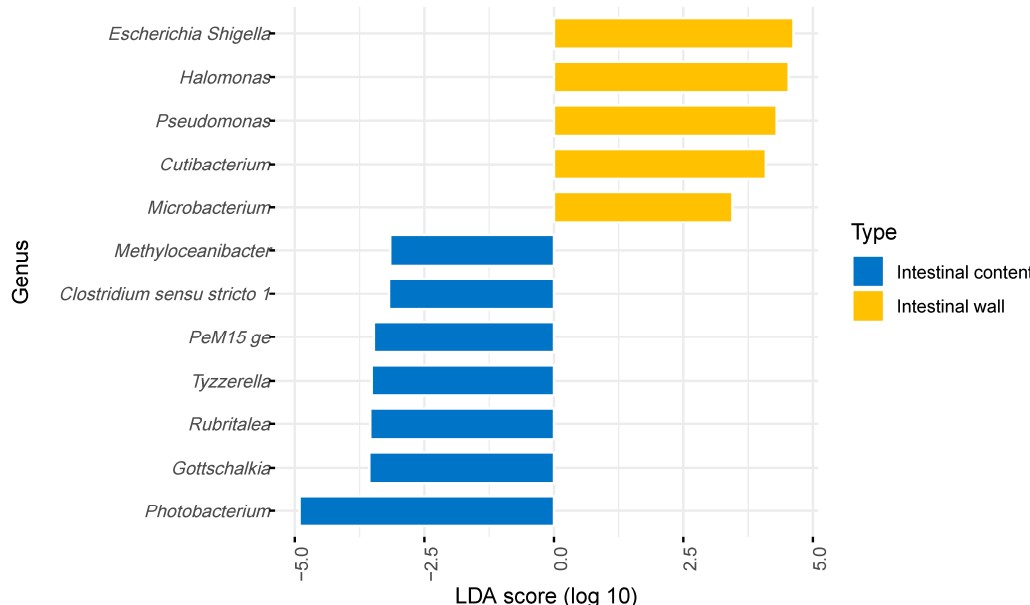

**Figure 4.** LEfSe analysis results showing differentially abundant genera in the intestinal content (n = 4) and intestinal wall (n = 4).

*Mycoplasma*, which was abundant in some of our samples, is one of the most dominant genera in the gut microbiome of Antarctic salmon [9,42]. However, it is unclear whether this group of bacteria is beneficial or pathogenic to the host. In Bozzi et al.'s study [13], *Mycoplasma* was associated with healthy Atlantic salmons, and their relative abundance was positively correlated with the body weight of their host. Based on metagenome-assembled genomes, Rasmussen et al. (2021) identified the functional potential of *Mycoplama* in de novo synthesis of arginine and ammonia detoxification suggesting the mutualistic relationships between *Mycoplasma* and its salmonid hosts. However, *Mycoplasma* has also been highlighted as a pathogen which may aid transmissible tumors in salmons [12,43].

*Escherichia*, and *Shigella*, and *Pseudomonas*, which are known to be pathogenic to fish species, were relatively more abundant in the intestinal walls compared with intestinal contents (Figure 4). Wu et al. (2021) [44] studied the gut microbiome of Nile Tilapia (*Oreochromis niloticus*) and found a higher relative abundance of *Escherichia* and *Shigella* in the intestinal mucosa than in intestinal contents, as in our study. In their study, the abundance of *Escherichia* and *Shigella* was negatively correlated with intestinal metabolites of the host. Many species belonging to *Pseudomonas* are pathogenic to fish species, causing ulcerative syndrome and hemorrhagic septicemia [45–47]. Higher relative abundance of *Escherichia* and *Shigella* and *Pseudomonas* in the intestinal wall suggests that they could be persistent members of the gut microbiome, which stay in the gut for longer than other (transient) microbial genera. The studied fish had no physical indications of disease; thus, it is unclear whether these bacterial groups are also harmful to Antarctic fish species. Notably, *Escherichia*, *Shigella*, and *Pseudomonas* encompass many bacterial species and sub-species; not all of them are pathogenic [48,49]. High-throughput sequencing methods which generate longer reads, such as Nanopore sequencing, could be helpful to further elucidate the relationship between these bacterial groups and their host.

The functional gene profile of the gut microbiome predicted by PICRUSt2 indicated a higher abundance of genes related to the degradation of chitin derivatives in the intestinal content compared with intestinal wall, which corresponded with the relative abundance of *Photobacterium* in the samples (Figure 5, Table S1). We also found a higher relative abundance of the genes related to nutrient degradation in the intestinal content compared with the intestinal wall, such as genes related to "starch degradation V", the "superpathway of N-acetylglucosamine, N-acetylmannosamine and N-acetylneuraminate degradation", "acetylene degradation", "purine ribonucleosides degradation", etc. In contrast, we found

a higher relative abundance of the genes related to biosynthesis in intestinal wall samples, such as the "superpathway of menaquinol-6, menaquinol-9, and menaquinol-10 biosynthesis", "superpathway of arginine and polyamine biosynthesis", "enterobactin biosynthesis", etc. The PICRUSt2 results suggests that the bacterial species found in intestinal walls may not all be pathogenic; some could be mutualistic, aiding the synthesis of essential metabolites.

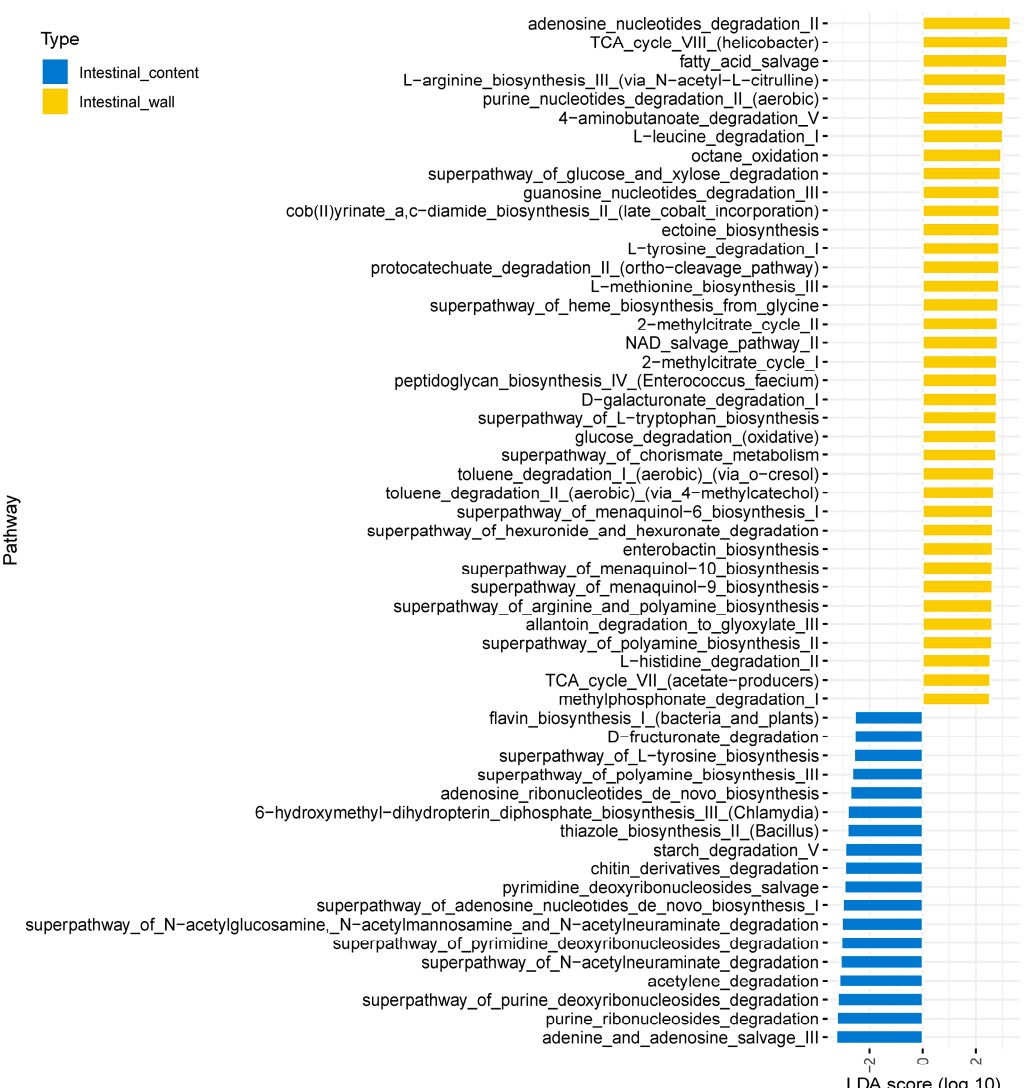

**Figure 5.** LEfSe analysis results showing differentially abundant (log-transformed LDA score > 2.5, $p < 0.05$) metabolic pathways (inferred by PICRUSt2) in the intestinal content (n = 4) and intestinal wall (n = 4).

## 4. Conclusions

In this study, for the first time, the gut microbiome of mackerel icefish has been characterized using next-generation sequencing. Overall, the gut microbiome of the Antarctic fish species, *C. gunnari*, was, to some extent, similar to other marine fish species, but different as well, lacking *Vibrio* and *Lactobacillus*. Although we found no difference in the alpha diversity between different sample types, we found distinctive microbial community compositions of the intestinal wall samples, suggesting niche separation in the gut environment. We identified potentially beneficial bacterial genera, such as *Photobacterium*, abundant in the intestinal content, which may contribute to the digestion of foods. However, potential pathogens such as *Escherichia*, *Shigella*, and *Pseudomonas*, were more abundant in the intestinal wall. The functional profile predicted by PICRUSt2 supported mutualistic relationships

between the gut microbiome and their host. In future studies, it would be necessary to collect intestinal wall samples together with intestinal content samples to understand the whole picture of the gut microbiome, because the microbial community structure and potential functions could vary depending on sample types.

**Supplementary Materials:** The following supporting information can be downloaded at: https://www.mdpi.com/article/10.3390/fishes8010013/s1, Table S1. LefSe analysis results showing differentially abundant (log transformed LDA score > 2.5, $p < 0.05$) metabolic pathways (inferred by PICRUSt2) in intestinal content (n = 4) and intestinal wall (n = 4). $p$-values were not adjusted; Figure S1: Photograph of a defrosted mackerel icefish, *Champsocephalus gunnari*; Figure S2: The use of spreaders for squeezing out the intestinal contents from the wall; Figure S3: Non-matric multidimensional plot based on the Bray–Curtis dissimilarity between samples. Samples are grouped by sex; Figure S4. Phylum composition of the *C. gunnari* gut microbiome. WI: whole intestine; IW: intestinal wall; IC: intestinal content.

**Author Contributions:** Conceptualization, H.S., D.-W.H. and J.-H.K.; Methodology, H.S., D.-W.H., S.L. and J.-H.K.; Formal Analysis, H.S.; Investigation, H.S., D.-W.H. and J.-H.K.; Writing—Original Draft Preparation, H.S.; Writing—Review and Editing, J.-H.K. All authors have read and agreed to the published version of the manuscript.

**Funding:** This research was supported by the Korea Polar Research Institute (PE22160), funded by the Ministry of Oceans and Fisheries. Additionally, this research was partly supported by the project titled "Development of potential antibiotic compounds using polar organism resources (20200610, KOPRI Grant PM22030)", funded by the Ministry of Oceans and Fisheries, Republic of Korea.

**Institutional Review Board Statement:** Not applicable.

**Data Availability Statement:** The raw fastq formatted sequence files have been archived in the NCBI SRA (sequence read archive) under project number of PRJNA857104.

**Acknowledgments:** The authors would like to thank the Jeong Il Corporation for supplying the fish samples.

**Conflicts of Interest:** The authors declare no conflict of interest.

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
