# Peer review of "Characterization of the Gut Microbiota of Mackerel Icefish, Champsocephalus gunnari"

_fishes, doi:10.3390/fishes8010013_

Round 1

Reviewer 1 Report (Previous Reviewer 1)

The authors did a great job adding alpha diversity, Lefse and PICRUSt analysis as well as more info on the methods and several references. Great job! However, there are a few minor revisions that need to be done:

1) Abstract and conclusion - not clear what the conclusions are. Please remove the last sentence in the abstract since it does not add anything. The sentence before that is not correct since you only used functional estimates (PICRUSt) to correlate what the gut microbes are doing but this was not the focus of your study and again are only estimates. Please revise and say something more to the point you found differences in the relative abundance of microbes between the intestinal wall and content and this is the first time the gut microbiome of mackeral icefish has been characterized using next-generation sequencing.

2) PICRUSt - need to add more info in the methods. For example, were KEGG pathways used and what did you use to correct for multiple testing (eg FDR) and what stats were used? 

3) Beta diversity figure 1 - this should go after figure 2.

4) Alpha diversity figure 2 - please include you found not significant differences.

5) Phyla abundance figure 3 - the legend says Genus and should say "Phyla". Also genera are in the legend and should be phyla. Please revise and this is a big mistake!

6) PICRUSt figure 6 - I think you wanted to say PICRUSt instead of "Lefse" (please revise). Also include p-values for each pathway. Also make longer vertically since text is squished.

7) All figures - should include sample size "(n=3-4)" so the reader knows the figures are not means. Also most figures have blurry text (except fig 2) so please export your figures with higher resolution(>300 dpi).

I agree with your points about Mycoplasma and Photobacteria. Great conversation! 

Author Response

Reviewer 2 Report (New Reviewer)

The manuscript with ID (fishes-2129965) by Song and coauthors has evaluated the gut microbiome of mackerel icefish, Champsocephalus gunnari, as a novel study. The present form of the manuscript is the revised version of the manuscript, in which the authors have modified and improved the quality of their work. 

In this sense, the authors have included new analyses (alpha diversity, and PICRUSt, and Lefse analysis) which improve the knowledge of the microbiota present in their samples.

I have only detected one minor error, which is the new Fig. 3 of this version. In this sense, as it is a Short Communication, I would recommend the authors remove this Figure (Line 168), and the references to the figure (line 160), and re-enumerate the following Figures and references to them in the text. This can be done when the authors receive the final proofread version of their manuscript. 

Thus, considering that the authors have submitted their work as a Short Communication, instead of a full article, in my opinion, the manuscript can be accepted in its present form.  

With that being said, I would recommend the authors to analyze more samples (whenever possible) in future analysis. 

Author Response

This manuscript is a resubmission of an earlier submission. The following is a list of the peer review reports and author responses from that submission.

Round 1

Reviewer 1 Report

Major Strengths

The manuscript titled “Characterization of the Gut Microbiota of Mackerel Icefish, Champsocephalus gunnari” focused on a wild species that has not been studied in terms of the bacteria in their gut. The strengths of the study included: analyses of bacteria using next generation sequencing, novel study of a wild fish that has not been analyzed for their gut bacteria and comparing different sections of the gut. The authors also did a good job removing non-bacteria taxa from their dataset, using the up to date SILVA database and subsampling that resulted in many sequences per sample.

Major Weaknesses

There were several weaknesses in the study. In general, the grammar needs to be improved and the study was very short and basic (not comprehensive) as it only looked at the gut microbiome of 8 wild fish. No metagenomics (PICRUSt) or Lefse analyses were performed and they did not include alpha diversity (Shannon diversity and Chao1 richness needs to be performed). Most NGS studies in my experience need a samples size of at least 6 per treatment, where this had 4.

The fish were also carried on a fishing boat for a while and possible contamination may have occurred. The authors did not include blank samples in the analysis to identify any potential contamination and no environmental samples of water or skin were compared. A higher sample size and environmental samples are recommended for future analyses.

L18 – I disagree, I think Mycoplasma and Pseudomonas are pathogenic and not beneficial. Please revise. See the Discussion in the two studies below and add these references to your Discussion. The refer to Mycoplasma associated with disease in zebrafish and Atlantic salmon:

Paquette, C. E., Kent, M. L., Buchner, C., Tanguay, R. L., Guillemin, K., Mason, T. J., et al. (2013). A retrospective study of the prevalence and classification of intestinal neoplasia in zebrafish (Danio rerio). Zebrafish 10, 228–236. doi: 10.1089/zeb.2012.0828

Huyben, D., Roehe, B. K., Bekaert, M., Ruyter, B., & Glencross, B. (2020). Dietary lipid: protein ratio and n-3 long-chain polyunsaturated fatty acids alters the gut microbiome of Atlantic salmon under hypoxic and normoxic conditions. Frontiers in Microbiology, 11, 589898.

Llewellyn, M. S., McGinnity, P., Dionne, M., Letourneau, J., Thonier, F., Carvalho, G. R., et al. (2016). The biogeography of the Atlantic salmon (Salmo salar) gut microbiome. ISME J. 10, 1280–1284. doi: 10.1038/ismej.2015.189

Minor Weaknesses

Throughout – grammar is very bad. Missing many adverbs and correct verbs. Please improve.

L13 – how large were these fish and were they wild caught? Please include.

L20-21 – not sure this sentence is needed. Please include common phyla found and alpha diversity. Also, was there a difference between sections?

L40 – add more references about salmon microbiome to indicate many studies have been done (see the above citations).

L43 – what other polar fishes have been investigated in terms of gut microbiome? Maybe it is better to focus on Southern hemisphere fish?

L47-49 – need to add references.

L61 – where is this subarea (closest land city?) and when were these samples collected? Please add

L62 – what temperature were they frozen and how long were they frozen before dissection? Please add

L64 – the weight, length and sex of the fish are critical info. Please move from supplemental material into the manuscript.

L74 – what do you mean by “total intestine”? Do you mean from the caeca to the anus (proximal and distal intestine) or are you including the esophagus, stomach and caeca? Please be more specific.

L79 – rinsed how many times? How long and at what speed was it homogenized? Were beads used?

L88 – how many PCR cycles were used, which polymerase and how were they indexed? What version of the MiSeq and what kit was used (300 cycles)?

L93 – did you use a cut off for sequence length and did you remove chimeras?

L98 – how did you analyze alpha diversity? Did you use common indices, such as Shannon and Chao1? You need to do this since diversity is very important metric of gut health. Please see the studies above for more info and there is also a function in Mothur.

L106 – what were the assumptions and how did you test for normal distribution?

L112 – mention this finding in your abstract

L115 – did you perform stats on sex differences? Please do this.

L124 – which specific fish species and livestock? Please include

Fig 1 – include x and y-axis numbers and labels. Include in the footer this is from wild C. gunnari. Also, most studies use the term “whole intestine” instead of digestive tract since tract include the stomach and caeca.

Fig. 2 – include description of DT, IC and IW in the footer. Also include fish name. Where is the phyla for Mycoplasma (Tenericutes or Mycoplasmatota)

Fig. 3 – see above comment.

L149 - Photobacterium has been found in the gut of marine fish (MacDonald et al., 1986) and in estuary

water environments (Austin, 2006), which suggests that the high abundance in the gut of farmed rainbow trout may derive from the fish meal ingredient or the river water supplied to the fish (Huyben et al, 2017). Please include this and references below:

Huyben, D., Nyman, A., Vidaković, A., Passoth, V., Moccia, R., Kiessling, A., ... & Lundh, T. (2017). Effects of dietary inclusion of the yeasts Saccharomyces cerevisiae and Wickerhamomyces anomalus on gut microbiota of rainbow trout. Aquaculture, 473, 528-537.

MacDonald, N., Stark, J., Austin, B., 1986. Bacterial microflora in the gastro-intestinal tract of Dover sole (Solea solea L.), with emphasis on the possible role of bacteria in the nutrition of the host. FEMS Microbiol. Lett. 35, 107–111.

Austin, B., 2006. The bacterial microflora of fish, revised. Sci. World J. 6, 931–945.

L171 – see comment on Mycoplasma in above Major Weaknesses section. I agree that Mycoplasma may have a mutualistic relationship in the gut of salmonids. The Huyben et al (2020) study above also found the largest Atlantic salmon had the highest abundance of Mycoplasma in the gut. However, both tumors and Mycoplasma have been found in the gut of older zebrafish and wild Atlantic salmon (Paquette et al., 2013; Llewellyn et al., 2016). The icefish were around 500g, making them adults, so they may fit into this same category.

L185 – Pseudomonas is not a common probiotic. Lactobacillus and Bacillus are. Pseudomonas is a large genus (191 in its family) so it is difficult to associate this with anything. There are several pathogenic species, such as P. aeruginosa, that causes disease (necrotising enterocolitis) in humans. Please remove sentences about probiotics and antibiotics

Fig 3. – were there no Vibrio or Lactobacillus? These are commonly found in fish (salmon and trout), so you should talk about the absence of these. Please include more genera (up to 20) in your figure since there are important ones missing. Most people show taxa with >1% abundance.

Results and Discussion – more references need to be included about common bacteria found in Antarctic fish as well as other fish, especially marine fishes, such as salmon and cod.

L196 – remove “beneficial”

L198 – there has been around 10 years of studies on the fish gut microbiome. I don’t think saying most work has been done on humans is correct anymore. Please remove. Please include common phyla found, differences between sections and alpha diversity (please analyze!)

Reviewer 2 Report

The obvious value of this work is that the composition of gastrointestinal microbiota has been studied on the example of a poorly studied and unusual (its members lack erythrocytes and the respiratory pigment haemoglobin) member of Antarctic fish species, the mackerel icefish (Champsocephalus gunnari). Such unique studies are of great interest and contribute greatly to the knowledge of native (autochthonous) host microbiota and features of development of bacterial diversity of intestinal communities of fish living in Antarctic waters. Numerous investigations of the gastrointestinal tract microbiota are mainly related to valuable commercial or aquaculture salmonid species.

Nevertheless, to the manuscript presented in this version there are significant questions that require clarification.

The main remark concerns the methodological part, to the method and amount of the material  - in my opinion, a small number of specimens, only 8 fish were studied. What does this have to do with? Also it is not indicated at what temperature and how long the fish was stored before it got to the lab? When exactly was the fishing done - month, year? It is known that fresh, live material must be used for genomic bacterial studies, either collected and stored in liquid nitrogen or refrigerated at -80C.

The Introduction section and Results with Discussion is too short and poorly detailed.

Line 74 Please explain why the whole intestine (Digestive track) was investigated? What was the purpose? In the M&M it says that 4 samples were taken for this, but then everywhere else it says only 3 samples. Where did another 1 sample go?

Line 99-100 says alpha diversity, but no results are given later.